# Principal Drivers of Fungal Communities Associated with Needles, Shoots, Roots and Adjacent Soil of *Pinus sylvestris*

**DOI:** 10.3390/jof8101112

**Published:** 2022-10-21

**Authors:** Diana Marčiulynienė, Adas Marčiulynas, Valeriia Mishcherikova, Jūratė Lynikienė, Artūras Gedminas, Iva Franic, Audrius Menkis

**Affiliations:** 1Institute of Forestry, Lithuanian Research Centre for Agriculture and Forestry, Liepų Str. 1, Girionys, 53101 Kaunas District, Lithuania; adas.marciulynas@lammc.lt (A.M.); valeriia.mischerikova@lammc.lt (V.M.); jurate.lynikiene@lammc.lt (J.L.); arturas.gedminas@lammc.lt (A.G.); 2Department of Southern Swedish Forest Research Centre, Swedish University of Agricultural Sciences, P.O. Box 190, SE-23422 Lomma, Sweden; iva.franic@slu.se; 3Department of Forest Mycology and Plant Pathology, Uppsala BioCenter, Swedish University of Agricultural Sciences, P.O. Box 7026, SE-75007 Uppsala, Sweden; audrius.menkis@slu.se

**Keywords:** Scots pine, fungal community, environmental conditions, needles, shoots, roots, soil

## Abstract

The plant- and soil-associated microbial communities are critical to plant health and their resilience to stressors, such as drought, pathogens, and pest outbreaks. A better understanding of the structure of microbial communities and how they are affected by different environmental factors is needed to predict and manage ecosystem responses to climate change. In this study, we carried out a country-wide analysis of fungal communities associated with *Pinus sylvestris* growing under different environmental conditions. Needle, shoot, root, mineral, and organic soil samples were collected at 30 sites. By interconnecting the high-throughput sequencing data, environmental variables, and soil chemical properties, we were able to identify key factors that drive the diversity and composition of fungal communities associated with *P. sylvestris*. The fungal species richness and community composition were also found to be highly dependent on the site and the substrate they colonize. The results demonstrated that different functional tissues and the rhizosphere soil of *P. sylvestris* are associated with diverse fungal communities, which are driven by a combination of climatic (temperature and precipitation) and edaphic factors (soil pH), and stand characteristics.

## 1. Introduction

Changing climate induces multiple abiotic and biotic risks to forests and forestry worldwide. Indeed, climate change affects the location, composition, structure and function of forests in many parts of the world, including high-latitude forests dominated by boreal species [1]. Nevertheless, forests may respond to climate change in various ways driven by local environmental conditions and the adaptive potential of the trees. Climate change is also changing the local environments, affecting local microbial communities and, thus, their metabolic activity and capacity [2]. Yet, climate, tree composition, or the distribution of the host tree do not always explain spatial patterns in the above- and below-ground microbial communities. This is important to consider because different parts of terrestrial plants, including forest trees, harbor several communities of microorganisms that form complex associations and play important roles that can determine the functioning and health of their hosts. There are many ways to tackle the challenges of climate change, including adaptation and mitigation strategies. However, little attention has been given to microbial adaptation [3]. Although microorganisms tend to adapt rapidly to changing environmental conditions [4], little is known about how these changes will feed back into plant–microbe interactions [5]. The overall effects of plant-associated microbes on host health and fitness are determined by a number of factors, including host and microbial genotypes, interactions within microbiota, and various abiotic factors [6]. The main abiotic factors that determine the diversity of tree microbiome include temperature, light, nutrient and water availability, and soil pH [7]. However, the role of abiotic factors in determining the microbiome structure of forest trees, especially in different habitats, is not well understood [8]. Tree-associated microbes occupy specific habitats, which are referred to as rhizosphere, rhizoplane, phyllosphere, or endosphere [9]. Microbes in these habitats can establish beneficial, neutral, or detrimental associations of varying intimacy with their hosts [10]. Fungi are among the dominant groups of plant-associated microorganisms as they play key roles in regulating plant health, maintaining interactions between plants and other organisms, and maintaining the functioning of entire ecosystems [11,12]. Maintaining functional tree-microbial interactions has been shown to be important for tree growth and development and for adaptation to the changing environment [13]. Given that different fungal species and functional groups are driven differently by climatic, nutritional, or biotic factors, it is expected that different fungal guilds may also react differently to changes in environmental conditions [14].

Some of the most complex microbiomes are found in soils [15]. Fungi are of particular interest as they represent a significant fraction of the soil microbial community and influence several ecosystem processes [16]. The abundance of fungal functional groups in the soil changes in response to forest disturbance and indicates a close interaction between the above-ground plant community and the below-ground soil biological community [14]. Moreover, structural and functional modifications in the soil microbiota have a crucial impact on above-ground ecosystems [17]. A global meta-analysis found that forest degradation reduces soil C and N content, increases soil pH, and increases C decomposition rates. The study also found a decrease in soil fungal biomass in disturbed sites but increased species diversity [18]. Changes in soil pH were also shown to be significantly correlated with changes in the fungal community composition in the soil [19]. Besides, the loss of host species and forest disturbance can also cause the direct loss of mycorrhizal and other rhizosphere fungi [20].

Several previous studies have examined the effects of individual disturbances on soil fungal composition, focusing on ectomycorrhizal (ECM) fungi [21,22,23,24] but largely excluding other fungal guilds with which they interact and synergize to perform multiple ecosystem processes [25,26]. ECM fungi are widely associated with trees in natural forests and plantations [27]. They are a key component of the forest soil microbiome, forming symbioses with ca. 60% of trees on Earth [28]. Their importance increases with the decrease of soil fertility and under harsh environmental conditions as they improve the host trees’ nutrition and stress resistance [20,29]. As previous studies have shown, fungal communities associated with tree roots show pronounced differences compared with the communities of the surrounding soil [7]. Differences between root and rhizosphere microbial communities have been found to be influenced by the seasonal production of roots and a variation in their activity [30] and stand age [31]. Root-associated fungal communities are also known to depend on environmental parameters [32,33,34], such as climatic conditions [35], soil chemical composition, especially pH, and the content of organic matter [36]. Soil and/or root-associated fungi can also be pathogenic [37] or interact as saprotrophs involved in nutrient recycling [38]. Pathogenic fungi can interfere with tree growth and alter the diversity and composition of the plant community [39].

Different above-ground functional tissues of forest trees can also host diverse communities of fungi [7]. For example, living needles of coniferous trees are associated with a diverse fungal community, and these fungi may have various effects on their host [40]. Such fungi commonly include endophytes and epiphytes that colonize the interior and exterior surfaces of living needles, respectively [41]. Besides, their abundances in needles vary depending on seasonal precipitation and temperature [42]. There are several factors that influence the composition of the foliar fungal community. These factors include the host species, nutrient content in the needles, needle age, air temperature, precipitation, and air pollution [43,44]. Recent studies also show that the fungal community of pine trees growing in a common environment show host species-specific structures [45]. The importance of endophytic and other wood-inhabiting fungi has also been emphasized. Endophytes are common in living tree tissues [46,47,48] and may influence many biotic processes in trees, but under certain conditions, they may shift to pathogenic or saprophytic lifestyles [7]. Therefore, they can also be involved in the early decomposition of deadwood, which is an important carbon stock in forest ecosystems [49,50] and an important habitat for many forest living organisms [51].

As fungi play a vital role in the structural and functional dynamics of terrestrial ecosystems [52], it is important to understand how the above- and below-ground fungal communities will respond to changing environmental conditions. Furthermore, exploring the impacts of local variables on the structure of fungal communities can help to better understand the ecological functions of fungi in different habitats. The focus of the present study was a Scots pine (*Pinus sylvestris* L.), which is one of the most economically and ecologically important tree species in Europe [53]. Still, climate change is likely to affect its survival and growth differently in different parts of its distribution range [54]. The aim of the present study was to study the diversity and composition of above- (needles, shoots) and below-ground (roots, soil) fungal communities associated with *P. sylvestris*, and to evaluate the impact of abiotic factors on these fungal communities. We hypothesized that above- and below-ground fungal communities will change in quantity and composition along with the changes in environmental factors (temperature, precipitation, soil chemical properties) and stand age. Furthermore, due to the importance of soil fungi for the establishment and function (nutrient uptake) of *P. sylvestris*, we also hypothesized that changes in soil fungal communities will have a greater impact on aboveground fungal communities than changes in aboveground fungal communities on fungi in the soil. Understanding these structural associations between above- and below-ground fungal communities will provide a relevant information about important functions linked to resilience and ecosystem functioning of *P. sylvestris* forests in the future. The results were also expected to provide new knowledge on overall fungal biodiversity associated with *P. sylvestris* that can contribute to the development of sustainable forest management strategies to effectively maintain forest biodiversity and different ecosystem services.

## 2. Materials and Methods

### 2.1. Study Sites and Sampling

The study included thirty sampling sites, each of which was 500 m^2^ in size, situated at least 100 m away from the forest edge, at least 15 km apart from each other, and followed the principal distribution of *P. sylvestris* in Lithuania (Figure 1). Information on the stand characteristics for each site is in Table 1. In each site, the health status of 30 trees was assessed by evaluating the proportion of dry branches, dechromation, and defoliation [55,56]. The sampling of needles, shoots, roots, and the rhizosphere soil was carried out between April and May 2019. At each site, five *P. sylvestris* trees were randomly selected, and from each tree, five branches that were ca. 17 m above the ground and growing out from the main stem were cut using a telescopic pruner. These branches were used to collone-year-old old, i.e., from the previous growing season (2018), needles and shoots. Four random needles were taken from each branch and put together, making one representative sample per tree consisting of 20 needles. For a sampling of shoots, needles were removed, and ca. 5 cm-long segment of a one-year-old shoot was randomly taken from each branch, making one representative sample per tree that consisted of five shoots. For a sampling of soil, the litter layer was removed, and three individual samples were taken down to 25 cm depth in the vicinity of five *P. sylvestris* trees using a 2.5 cm diameter soil core, which was carefully cleaned between individual samples. The positions of sampled soil cores were at the northern, northeastern, and southwestern sides of each tree and within a distance of 0.25 m of the tree trunk. After the collection, organic and mineral layers were separated and sieved (mesh size 2 mm × 2 mm) to remove larger particles and roots, and each layer from different samples pooled together. For soil chemical analyses, five additional sub-samples per site were taken in the vicinity of *P. sylvestris* trees and processed as described above. Soil chemical analyses were carried out at the Agrochemical Research Laboratory of the Lithuanian Research Centre for Agriculture and Forestry, Kaunas, Lithuania. Fine roots (lateral with root tips) were excavated within a distance of 0.5 m from the stem of five *P. sylvestris* trees and separated from the soil and other particles. All equipment was thoroughly cleaned between individual samples. After the collection, individual needle, shoot, root, and soil samples were placed in sterile plastic bags, labeled, placed on dry ice, the same day transported to the laboratory, and stored at −20 °C until further processing. In total, there were 150 needles, 150 shoots, 150 roots, and 180 soil samples collected. Climate data were obtained from the nearest meteorological stations.

### 2.2. DNA Work

Prior to DNA extraction, root samples were carefully washed in sterile water to remove any of the remaining soil and cut into ca. 5 mm-long segments. All needle, shoot, root, and soil samples were freeze-dried for 48h using Labconco FreeZone Benchtop Freeze Dryer (Cole-Parmer, Vernon Hills, IL, USA). Then, ca. 0.5 g of freeze-dried material was taken from each sample and individually ground to a fine powder using a Fast prep shaker (Montigny-le-Bretonneux, France). Approximately 30 mg of this powder per sample was used for DNA extraction. The total DNA was extracted using the CTAB method as described by Marčiulynas et al. [6]. Extracted DNA was quantified using a NanoDrop™ One spectrophotometer (Thermo Scientific, Rochester, NY, USA). PCR amplification of ITS rRNA region using barcoded primers gITS7 [57] and barcoded primers ITS4 [58] was performed according to the protocol of [59]. Within the same site, samples of the same substrate (needles, shoots, roots, and the soil) were amplified using primers with the same barcode, resulting in these samples pooling together following PCR. Individual PRRs were performed to increase the representativeness of each site. PCR amplification was performed in 50 μL reactions using an Applied Biosystems 2720 thermal cycler (Foster City, CA, USA). The PCR program started with an initial denaturation step at 94 °C for 5 min, followed by 30 cycles of 94 °C for 30 s, and annealing at 56 °C for 30 s and 72 °C for 30 s, followed by a final extension step at 72 °C for 7 min. The PCR products were assessed using gel electrophoresis on 1% agarose gels stained with GelRed (Biotium, Fremont, CA, USA). The PCR products were purified using 3 M sodium acetate (pH 5.2) (Applichem GmbH, Darmstadt, Germany) and 96% ethanol mixture (1:25). After quantification of PCR products using a Qubit fluorometer 4.0 (Life Technologies, Stockholm, Sweden), samples were pooled in an equimolar mix and used for PacBio sequencing using two SMRT cells at the SciLifeLab in Uppsala, Sweden.

### 2.3. Bioinformatics

Sequence quality control and clustering were performed using the SCATA NGS sequencing pipeline (http://scata.mykopat.slu.se, accessed on 14 April 2021). Quality filtering was done by removing short sequences (<200 bp), sequences with low read quality (Q < 20), primer dimers, and homopolymers, which were collapsed to 3 base pairs (bp) before clustering. Sequences that did not have a tag or primer were excluded, but information about linking the sequence to the sample was stored as metadata. The sequences were clustered into different OTUs using single-linker clustering based on 98% similarity. For each cluster, the most common genotype (real read) was used to represent each OTU. For clusters with only two sequences, a consensus sequence was created. Fungal OTUs were taxonomically identified using both the RDP classifier available at https://pyro.cme.msu.edu/index.jsp, accessed on 18 May 2021 (Centre for Microbial Ecology, Michigan State University, East Lansing, MI, USA) and the GenBank (NCBI) database using Blastn algorithm. The criteria used for identification were: sequence coverage >80%, similarity to species level 98–100%, and similarity to genus level 94–97%. Representative sequences of fungal non-singletons as the Targeted Locus Study project have been deposited in GenBank under accession number KFVY00000000. Taxonomical information was also associated with each cluster using the SH mapping feature using the UNITE database (https://unite.ut.ee/analysis.php, accessed on 26 July 2021). Fungal functional groups were assigned using the FUNGuild database (https://github.com/UMNFuN/FUNGuild, accessed on 26 July 2021) according to Nguyen et al. [60] and Tedersoo et al. [61].

### 2.4. Statistical Analyses

The effects of the substrate, environmental variables, and soil characteristics at a site level on OTU richness were assessed using generalized linear mixed-effect models using the glmmTMB function from the glmmTMB package [62]. Correlation between predictor variables was assessed using the cor function in R [63]. When a correlation coefficient between two variables was higher than 0.7, only one variable was selected to be used in the final model. A final model contained the following variables: tree composition, age, defoliation (stand characteristics), air temperature, precipitation (environmental variables), and soil pH, P_2_O_5_, K_2_O, and Ca (soil variables). Interactions between each variable and the substrate were included in the model to assess if the effects of a variable are consistent across all substrates. The site was included in the model as a random factor. All continuous variables were scaled using the function scale in R [63]. Truncated Poisson distribution for errors was assumed as all samples contained fungi. Model predictions were calculated using ggpredict from the package ggeffects [64] in R and plotted using the ggplot function from the ggplot2 package [65]. Non-metric multidimensional scaling (NMDS) based on Bray-Curtis dissimilarity was used to visualize fungal community structure among different substrates (needles, shoots, roots, and the mineral and organic soil). Prior to NMDS analysis, which was completed using a metaMDS function from the vegan package [66] in R, fungal reads were rarefied to 500 reads per sample to remove the effect of a differential sequencing depth. This eliminated 39 out of 148 samples from the data set. We used the permutational multivariate analysis of variance (PERMANOVA) with the Bray–Curtis distance metric to assess the significance of community similarity as a function of the substrate. PERMANOVA was done using the adonis2 function from the vegan package [66] in R. Pairwise comparisons between substrate levels were performed using pairwise.perm.manova function from the RVAideMemoire [67] package in R with 999 permutations. The Benjamini & Hochberg method was used for the adjustment of *p*-values for multiple comparisons [68]. The Shannon diversity index and qualitative Sørensen similarity index were used to characterize the diversity of fungal communities [69,70]. The nonparametric Mann–Whitney test in Minitab v. 18.1 (State College, PA, USA) was used to test if the Shannon diversity index among different samples was statistically similar or not.

## 3. Results

A total of 277,050 reads passing quality control were clustered into 2914 non-singleton OTUs (Table 2), while singletons were removed. Among the non-singletons, the most abundant were fungi, with 2602 (89.3%) OTUs (Table 2). Non-fungal OTU were removed from further analyses. The Shannon diversity index of fungal OTUs varied between 1.6 and 5.1 for different samples and sites, with the lowest found in root samples and the highest in soil samples (Table 2). The Shannon diversity index of fungal OTUs was significantly higher in needles than in roots, in shoots than in roots, and in the mineral and organic soil than in roots (*p* < 0.05) (Figure 2). In a similar comparison, other samples did not differ significantly from each other (Figure 2).

Ascomycota was the most abundant phylum in all substrates and sites, accounting for 75.4% of all sequences, followed by Basidiomycota (21.4%), Zygomycota (2.9%), Chytridiomycota (0.2%), and Glomeromycota (<0.1%) (Appendix A). Venn diagram revealed the common and unique fungal OTUs among different substrates (Figure 3). There were 109 (4.21%) OTUs that were shared among all samples. Soil harbored the highest percentage of unique OTUs (32.1%, 832 OTUs), followed by roots (8.9%, 232 OTUs), shoots (7.1%, 183), and needles (5.8%, 150 OTUs). A high percentage of unique OTUs in the soil indicates substrate-specific fungal communities. Species accumulation curves of different sample types (needles, shoots, roots, and the soil) were approaching the asymptote, showing the sequencing depth was largely sufficient and that nearly all OTUs were detected (Figure 4).

At the class level, 19 dominant classes were identified, accounting for 99.4% of all high-quality reads (Figure 5). The distribution and relative abundance of fungal classes varied among different substrates. The most dominant fungal classes in shoot samples were Eurotyomycetes (33.0%), Dothideomycetes (31.0%), and Leotiomycetes (10.9%), in needle samples—Dothideomycetes (49.4%) and Eurotyomycetes (15.1%), in roots samples—Agaricomycetes (31.6%), Leotiomycetes (27.1%), and Mucoromycotina_Incertae sedis (9.6%), while in soil samples—Agaricomycetes (21.1%), Dothideomycetes (19.5%), Leotiomycetes (11.4%), Archaeorhizomycetes (11.3%), and Eurotyomycetes (10.0%).

Identification at least to genus level was possible for 1396 (53.7%) out of 2602 fungal OTUs. Information on the 30 most common fungal OTUs representing 50.0% of all high-quality reads is in Table 3. Among these, nine OTUs representing 14.6% of all high-quality reads could not be identified to the species or genus level. The most common OTUs in all the samples were *Dothideomycetes* sp. 5208_5 (6.9%), Unidentified sp. 5208_1 (5.5%), *Archaeorhizomyces* sp. 5208_0 (4.3%), Unidentified sp. 5208_2 (3.9%) and *Helotiales* sp. 5208_17 (2.5%). The most common OTUs in the needle samples were Unidentified sp. 5208_5 (20.4%), *Phacidium lacerum* (4.5%), Unidentified sp. 5208_2 (4.4%), Unidentified sp. 5208_1 (4.3%), and *Sydowia polyspora* (3.4%). The most common OTUs in the shoot samples were Unidentified sp. 5208_1 (12.8%), Unidentified sp. 5208_2 (7.1%), *Rhinocladiella* sp. 5208_3 (6.4%), *Helotiales* sp. 5208_17 (6.1%). The most common OTUs in root samples were *Phialocephala fortinii* (11.1%), *Mycena cinerella* (8.9%), *Archaeorhizomyces* sp. 5208_0 (8.4%), and *Penicillium camemberti* (5.3%). The most common OTUs in the soil samples (organic and mineral soil combined) were *Archaeorhizomyces* sp. 5208_0 (9.0%), Unidentified sp. 5208_5 (2.6%), *Umbelopsis nana* (2.5%), and *Malassezia restricta* (2.4%).

Among the most common fungal OTUs, there were several for which the relative abundance varied significantly among the 30 sampling sites and substrates (Table 3). For example, in shoots, these were *Leptosphaeria* sp. 5208_64, *Lecania naegelii,* and *Rhinocladiella* sp. 5208_3. In roots, these were *P. camemberti*, *Mucor abundans*, *M. hiemalis,* and *Armillaria ostoyae*.

The assessment of fungal functional groups showed that most fungal OTUs could not be assigned to any functional group, primarily because these could not be identified to the genus or species level. Among the remaining OTUs, 7.6–10.7% per site constituted plant pathogenic fungi, 9.5–14.0%—saprotrophs, and 2.0–14.9% endophytic fungi (Figure 6). In terms of sequence reads, in different samples, fungi of the unknown functional group constituted 54.6–81.4% of reads, pathogens—6.5–12.4%, saprotrophs–6.1–11.7%, and endophytic fungi—2.0–20.4% (Figure 6). Consequently, the distribution of fungal functional groups substantially differed when compared between OTUs and sequence reads.

Generalized linear mixed effect models showed that species richness of fungal OTUs associated with roots increased with the increase of stand age while the opposite was the case for fungi in needles and organic soil (Chisq = 89, df = 4, *p* < 0.05). The stand age did not have an effect on the fungal species richness in shoots or the mineral soil (Figure 7A). In both organic and mineral soil, the species richness increased with the increase of defoliation, while the opposite relationship was in needles, shoots, and roots (Chisq = 133, df = 4, *p* < 0.05) (Figure 7B). Climatic variables also showed a significant effect on the species richness that varied depending on the substrate (Chisq = 95, df = 4, *p* < 0.05 and Chisq = 252, df = 4, *p* < 0.05 for temperature and precipitation, respectively) (Figure 7C,D). In needles, the fungal species richness increased with the increase in temperatures and precipitation. The opposite was for shoots and soil fungi, i.e., the species richness decreased with the increase of temperature and precipitation. In roots, the fungal species richness increased with the increase in temperature, but no effect of precipitation was found (Figure 7C,D).

The soil parameters significantly affected the species richness (Figure 8). The increasing concentration of P_2_O_5_ increased the fungal species richness in needles but decreased in the shoots and mineral soil (Chisq = 88, df = 4, *p* < 0.05) (Figure 8A). For the roots and organic soil, the concentration of P_2_O_5_ had no significant effect (Figure 8A). The increasing concentration of K_2_O increased the fungal species richness in shoots but decreased in other substrates, i.e., needles, roots, minerals, and organic soil (Chisq = 24, df = 4, *p* < 0.05) (Figure 8B). Soil pH had a similar effect on fungal species richness in the needles, minerals, and organic soil, i.e., species richness decreased with the increase in pH (Chisq = 127, df = 4, *p* < 0.05). The opposite was in the shoots, while the fungal species richness in roots was generally unaffected (Figure 8C). In all substrates, the fungal species richness increased with the increase of Ca concentration (Chisq = 23, df = 4, *p* < 0.05) (Figure 8D).

The type of substrate (needles, shoots, roots, or soil) had a strong effect on the composition of the fungal communities as the fungal communities associated with different substrates differed significantly from each other (PERMANOVA: df = 4, SumOfSqs = 12.554, R^2^ = 0.32793, F = 12.687, *p* < 0.001) (Table 4). An exception was the fungal communities in organic and mineral soil, which were similar (*p* > 0.05). Similarly, differences in the fungal communities among different substrates were also indicated by the NMDS of Bray–Curtis dissimilarity based on the OTU-level abundance (stress = 0.15) (Figure 9). PERMANOVA has shown that the fungal community composition varied among study sites, which explained almost as much variation in the fungal community composition as it was explained by the substrate (df = 29, SumOfSqs = 8.055, R^2^ = 0.210, F = 1.179, *p* < 0.01).

## 4. Discussion

As climate change is expected to affect the natural distribution of forest tree species, entire forest ecosystems are likely to be subjected to major changes to adapt to new conditions. However, it is less clear whether and how global fungal biodiversity will respond to changes in the distribution range of the host trees. In the present study, to get a deeper understanding of these questions, the impact of abiotic factors on the above- and below-ground diversity and composition of fungal communities were studied in 30 *P. sylvestris* sites. Using the high-throughput sequencing data, different environmental variables, and soil chemical properties, we identified key factors that drive the diversity and composition of fungal communities associated with *P. sylvestris*. Moreover, by linking the fungal community data and different substrates, we have demonstrated that the fungal species richness and community composition depend on the substrate they colonize. Such patterns of fungal specificity for the substrate have already been shown in *Picea abies* stands [71]. Besides, the fungal species richness can be different in the same tissues depending on their life stage, e.g., living or dead wood [49]. In addition, we have shown that the fungal species richness is highly dependent on the site. The site-specific composition of fungal communities was shown previously for soil, root, or foliage samples [72,73,74,75]. Overall, our results demonstrated that different functional tissues and the rhizosphere soil of *P. sylvestris* are associated with diverse fungal communities driven by a combination of climatic and edaphic factors, thereby providing insights into possible ecological responses of fungal communities to climate change in northern Europe.

Climate is one of the main factors influencing fungal development, either directly or indirectly, by triggering tree responses [12]. However, the diversity of fungal communities can be influenced by a variety of factors, including tree species composition, stand age, habitat conditions, and edaphic factors [76]. Despite many studies on interactions between soil fungi and their hosts, the interactions between fungi and above- and below-ground tree traits are less studied. It is emphasized that when assessing the interaction of above- and below-ground communities, it is important to identify the dynamics of soil microbial communities that reflect changes in forest age and soil properties [77]. Our findings suggest that the fungal species richness in needles, organic soil, and roots of *P. sylvestris* change with the stand age. In agreement with other studies [78], the greatest positive effect of stand age was on fungal species richness in roots. Such results could be due to the accumulation of fungal species over time as well as due to changes in nutrient supply by host trees [71,78]. Several studies have also reported a strong effect of tree age on fungal communities, as the litter quality and understory vegetation in young stands differ markedly from older stands [79]. It was shown that fungal communities in roots might be specifically shaped at the fine scale, but this may disappear when averaged across an entire landscape, obscuring the specific environmental conditions and species interactions that drive fungal diversity [80]. Indeed, the assembly of root-associated fungi has been shown to be regulated by a wide range of spatial and temporal variables [81].

Fungal communities are dynamic components of terrestrial ecosystems and exhibit temporal and spatial variation [77]. Across the range of a single host tree species, the fungal community may change in response to climatic factors [82]. It is also known that moisture and temperature influence the growth of fungi and that characteristic weather conditions favorable for fungi may be used to predict their abundance and richness in habitats with different climatic conditions [83]. Although climatic factors such as temperature and precipitation were found to promote fungal richness globally [84], our results suggest that the richness of fungal OTUs in the organic soil layer decreases with the increase in temperature and precipitation. Temperature is one of the main factors determining soil microorganisms’ activity [85]. It is known that temperature enhances microbial metabolic activity, potentially leading to an accelerated litter decomposition rate [86]. Low temperature is considered to be one of the dominant forces protecting soil C from decomposition [87]. However, it seems likely that soil organic C will decrease with increasing temperature due to climate change [88]. With an increase in temperature, evaporation is enhanced, and plant productivity is reduced, ultimately resulting in a decrease in soil organic carbon input [89]. Finally, shifts in the amount of above- and below-ground organic matter inputs to the soil may also shape the composition and activity of fungal communities [89,90]. Studies by Dang et al. [77] have also shown that the abundance of dominant fungal communities is significantly correlated with organic C, total N, C: N, available N, and available P, indicating the dependence of these microbes on soil nutrients. Therefore, it could be assumed that as the temperature increases and the C content decreases, the abundance of microbial communities is prone to decrease. Yet, large knowledge gaps remain that feed uncertainty around the temperature and sensitivity of soil microbial processes [91].

Recent studies have shown that the decrease in the abundance of saprotrophic fungi may have been a direct result of forest disturbance or an indirect result of changes in soil pH and soil P [92]. Furthermore, the dominant P-solubilizing saprotrophic fungi are replaced by diverse facultative pathogenic fungi with weaker C decomposition ability. These changes potentially indicate a shift from soil phosphate limitation to carbon limitation following deforestation [92].

Some belowground properties, including plant richness and plant diversity, total carbon, total nitrogen, soil pH, and nutrient content, change simultaneously with changes in the aboveground structure [77]. Furthermore, these indices eventually influence the function and structure of the soil microbial community [77]. Our study has shown that *P. sylvestris* fungal communities in the soil, roots, and shoots negatively respond to the increase in soil pH. The effects of soil pH on the soil microbial community assembly occur through the availability of nutrients and carbon and the solubility of metals, which are strongly influenced by changes in soil pH concentration [19]. Previous studies also confirm that soils with lower pH concentrations increase the rate of fungal growth in contrast to soils with higher pH [93]. Decreased fungal species richness in forest soils with increasing pH is also associated with decreased availability of carbon [93,94]. The increase in soil pH also reduces the rhizosphere priming effect, which leads to an increase in the accumulation of carbon in the soil, which may also be one of the reasons for the decrease in fungal richness in the soil [94]. It is also important to mention that soil pH is closely related to vegetation and strongly influences the degree of ectomycorrhizal colonization. In particular, it has been observed that a decrease in pH results in a shift in vegetation, leading to an increase in ectomycorrhizal fungi [95].

In the present study, the most common fungal phyla among different *P. sylvestris* substrates were Ascomycota and Basidiomycota, which are known to dominate among fungi colonizing soils and terrestrial plant tissues [7,11,96,97]. Interestingly, *P. sylvestris* needles showed a higher relative abundance of fungal OTUs belonging to Helotiales than Dothideales, confirming the hypothesis that Helotiales are dominated in gymnosperms and Dothideales in angiosperms [47]. The major differences between the above- and below-ground communities were the dominance of the Dothideomycetes in shoots and needles, while *Archaeorhizomyces* dominated in roots and soil samples. This is not surprising as representatives of both *Dothideomycetes* and *Archaeorhizomyces* were among the most commonly found fungal classes in pine forests [98,99,100,101].

Fungi colonizing different *P. sylvestris* tissues accounted for only a fraction of all fungal OTUs. As a result, communities of soil fungi had higher diversity than those inhabiting the aboveground parts, and in the organic soil, the abundance of fungi was higher than in the mineral soil. Numerous studies indicate that soil is the primary source of microbiomes in terrestrial habitats [102]. Since resource availability is a key factor regulating biodiversity and ecosystem functioning [103], it could be that soil harbors a greater diversity of microbes by providing more enriched or easily accessible resources and being an ecotone for both above- and below-ground communities. It has previously been shown that plant species can modify the soil environment and support rhizosphere microbes, which in turn can provide feedback that promotes plant health and growth [104]. The study by Bulgarelli et al. [105] on *Pinus ponderosa* sampled from several locations around the globe revealed minimal geographic differentiation of phyllosphere bacterial communities, which supports the concept that the host plant species is a determinant for the structure of the phyllosphere community. In comparison to other tissues and the soil, *P. sylvestris* needles were more often colonized by plant pathogens. It is important to emphasize that foliar pathogens, while infecting and causing the disease to the needles or leaves, can also, especially at high infection levels, cause growth reduction, increase susceptibility to biotic and abiotic stresses, and in severe cases, cause tree mortality [106,107]. This may also suggest that fungal communities in needles and roots, which are the primary functional tissues of trees, respond faster to biotic and abiotic stresses, while in the soil and shoots, these changes are slower [108].

Among soil and root-associated fungi, ectomycorrhizal fungi (ECM) represent a key component needed for the successful establishment and growth of *P. sylvestris* trees. Among these, Suilloid fungi are of key importance for pines, especially those of the *Suillus* and *Rhizopogon* genera [109]. Despite the key importance of these ECM fungi, they occupied only 0.5% of the total diversity in the soil and 0.1% in the roots of investigated samples. Suilloid fungi represent species of early succession and, with the age of the stand, are often replaced by other ECM fungi, which leads to the fact that they become less common [110,111]. Besides, a higher proportion of this group of fungi is usually found after forest disturbance [109]. For example, *Rhizopogon* species were found to dominate ECM communities associated with pine roots in post-fire sites [112]. Among other more frequently detected fungi was *Umbelopsis* nana, which accounted for 2.7% of the total fungal diversity. Members of the genus *Umbelopsis* are common soil fungi that are found in forest ecosystems around the world [113,114]. This species appears to be an indicator of the mineral soil habitats in many northern forests [115]. Studies in Swedish clear-cut *P. sylvestris* forests also revealed that *U. nana* was one of the most common species of soil fungi [116]. In previous studies, members of this genus were also detected in the roots of *P. sylvestris* [101], *P. abies* [117], *Picea mariana* [115], *Pseudotsuga menziesii, Pinus ponderosa* [114], and *Quercus* sp. [118]. *Umbelopsis* fungi are also known to synthesize polyunsaturated fatty acids, which are important in plant stress defense [119] and can be important in the process of climate change due to the increase of stressors. *Phacidium lacerum*, which was frequently detected in the soil and shoot samples, is widely distributed throughout Europe, where it commonly occurs on *P. sylvestris* [120], but it is not known to cause the disease on this host [121]. However, it was recently reported to be the causal agent of postharvest rot in apples and pears [122]. Therefore, the pathogenic nature of pines should not be excluded. *Sydowia polyspora*, often found in needles and shoots, is commonly associated with conifers worldwide and is considered a pathogen on several hosts [123]. Moreover, Cleary et al. [124] recently found *S. polyspora* in asymptomatic seeds of several *Pinus* spp. obtained across Europe and North America, including *P. pinaster*, *P. radiata*, *P. strobus*, *P. sylvestris*, *P. mugo,* and *P. pinea*. Fungus is also vectored by insects [125] and very often reported as an endophyte or a saprophyte [126]. The endophytic way of life of pathogenic fungi can be a starting point at an early stage of the infection process before transitioning to a pathogenic lifestyle [127].

## 5. Conclusions

Our study has shown that *P. sylvestris* forests in Lithuania are associated with a high diversity of above- and below-ground fungal communities, which are strongly influenced by interrelated environmental factors and stand characteristics. Patterns of fungal community composition were strongly associated with particular substrates, showing that they provide different ecological niches that are preferred by different fungal species. To maintain the availability and suitability of ecological niches for different fungal species, it is essential to maintain an appropriate soil pH and the availability of soil nutrients. Disturbances related to climate change are likely to affect these important factors and, thus, may indirectly affect the structure and diversity of fungal communities. On the practical side, forest management and planning in the future will have to adapt to changing climatic conditions to ensure ecosystem services through the introduction of new management approaches. This is particularly important for maintaining soil health. In the future, the stands with other, possibly non-native, tree species, which better tolerate climate change, may also become unavoidable to maintain the diversity of microorganisms and ensure the growth and sustainability of forest stands.

## Figures and Tables

**Figure 1 jof-08-01112-f001:**
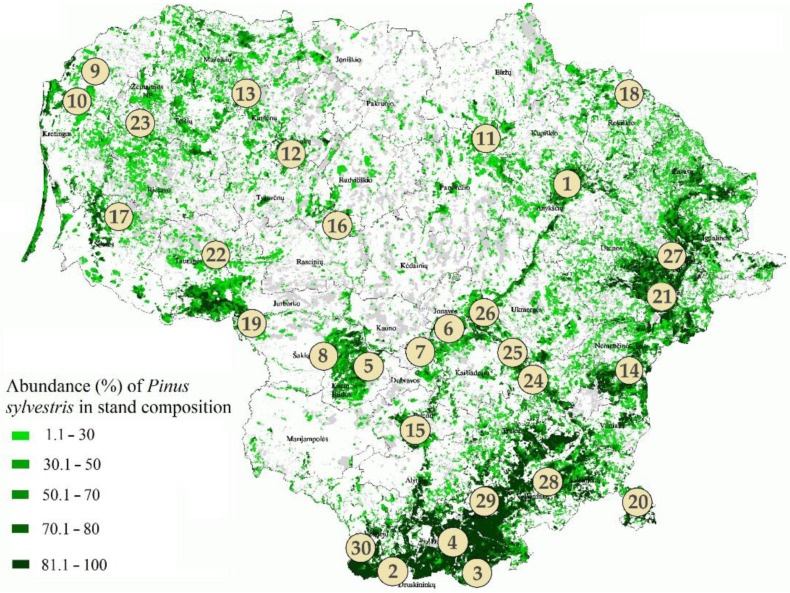
Map of Lithuania showing the sampling sites in *Pinus sylvestris* forest stands. The green color on the map shows the distribution of *P. sylvestris* stands and its abundance (%) in different areas.

**Figure 2 jof-08-01112-f002:**
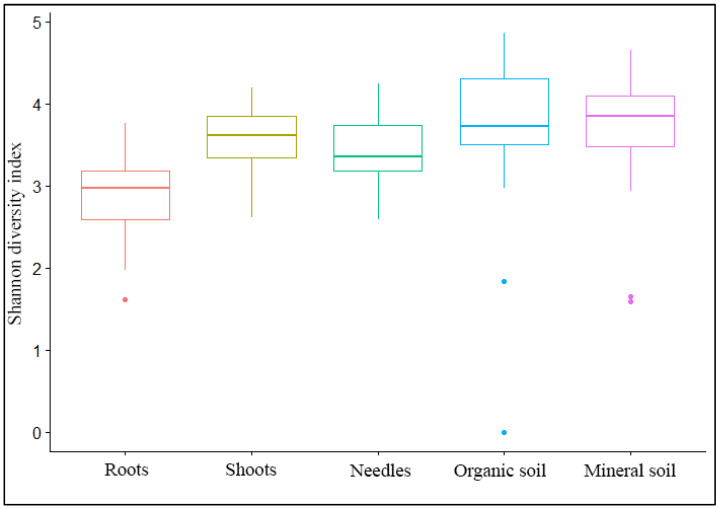
Differences in the Shannon diversity index of fungal OTUs among different substrates. In the Mann–Whitney test, *p*-values were: roots vs. shoots *p* < 0.05, roots vs. needles *p* < 0.05, roots vs. organic soil *p* < 0.05, root vs. mineral soil *p* < 0.05, shoot vs. needles *p* = 0.97, shoots vs. organic soil *p* = 0.06, shoots vs. mineral soil *p* = 0.23, needles vs. organic soil *p* = 0.06, needles vs. mineral soil *p* = 0.06, and organic soil vs. mineral soil *p* = 0.99.

**Figure 3 jof-08-01112-f003:**
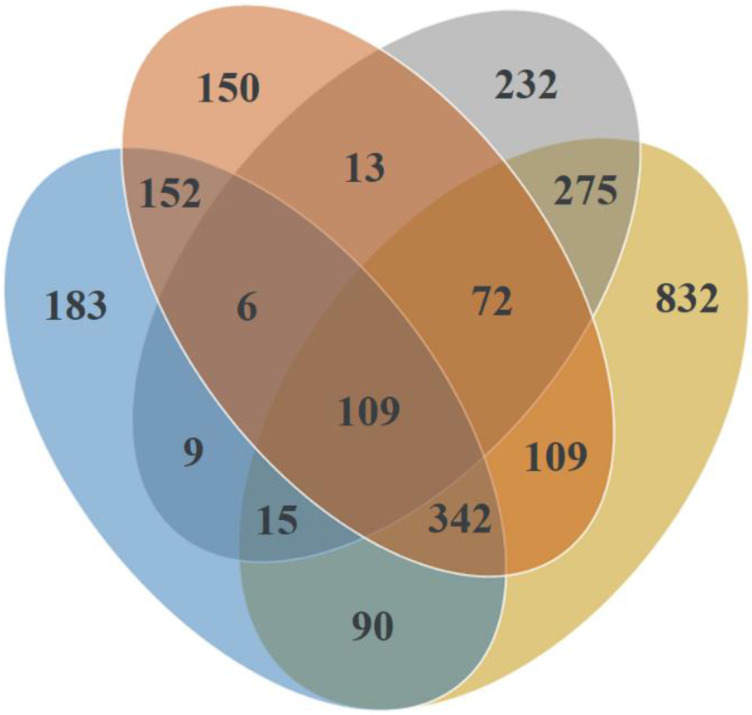
Venn diagram showing the diversity and overlap of fungal taxa in different sample types from *P. sylvestris* stands. Different colors represent different substrates: Yellow—Soil (organic and mineral combined), Orange—Needles, Blue—Shoots, and Gray—Roots.

**Figure 4 jof-08-01112-f004:**
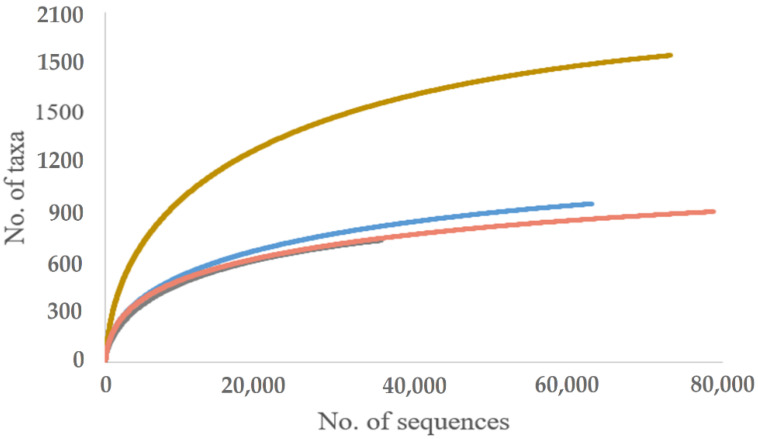
Species accumulation curves showing the relationship between the cumulative number of fungal OTUs and the number of ITS rRNA sequences shown as Yellow—Soil, Orange—Needles, Blue—Shoots, and Gray—Roots.

**Figure 5 jof-08-01112-f005:**
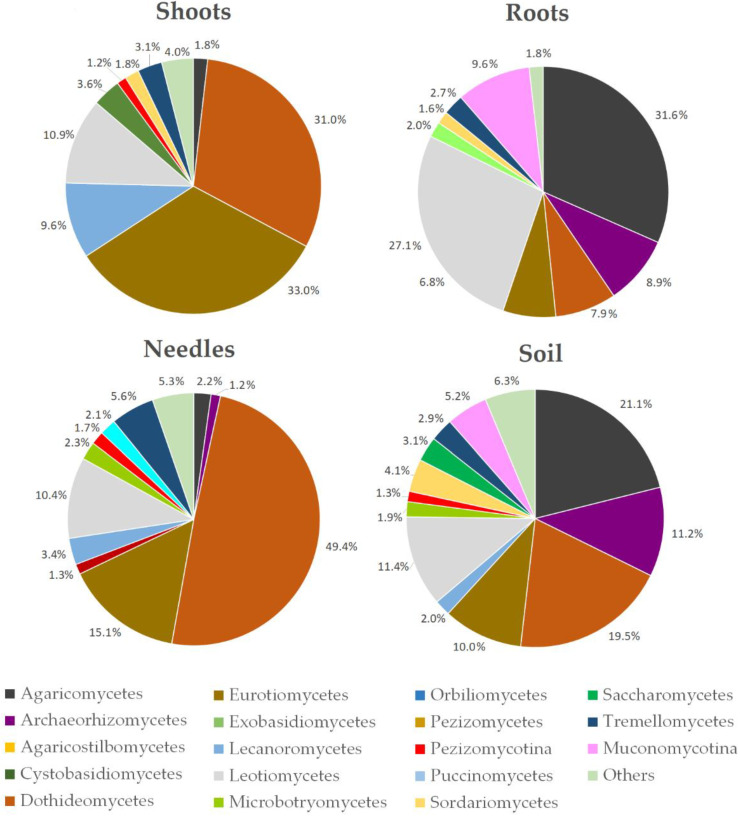
Relative abundance of fungal classes in needles, shoots, roots, and the soil of *Pinus sylvestris*. Other represented fungal classes with a relative abundance of <1%. The data from the 30 different sample sites are combined.

**Figure 6 jof-08-01112-f006:**
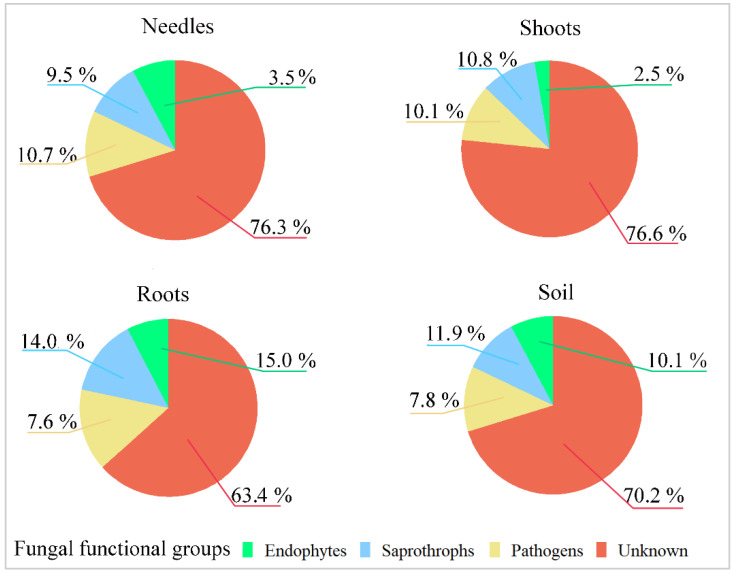
Fungal functional groups in the needle, shoot, root, and soil samples from the 30 sampling sites of *Pinus sylvestris* are shown as a relative abundance (%) of the total number of fungal taxa.

**Figure 7 jof-08-01112-f007:**
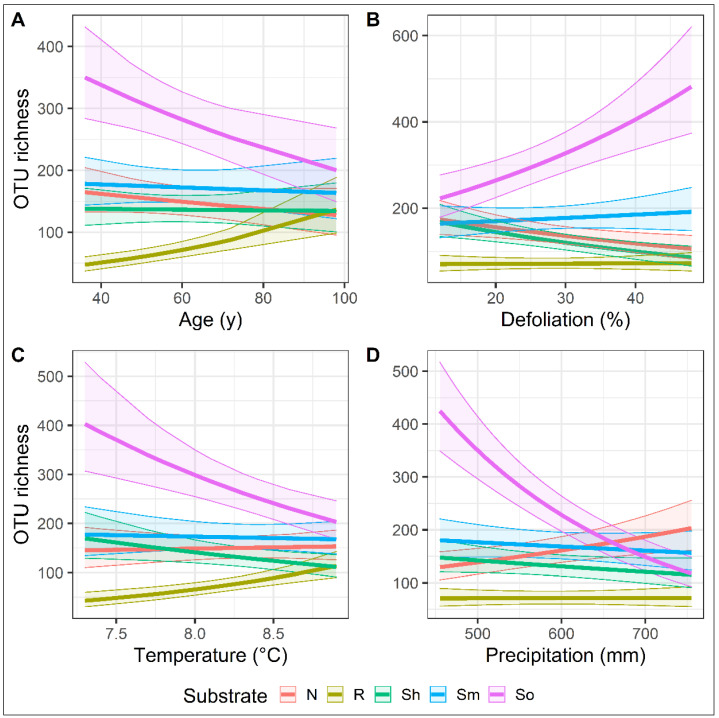
The relationship between the species richness of fungal OTUs in different substrates (needles, shoots, roots, mineral, and organic soil) of *Pinus sylvestris* and different stand (**A**,**B**) and climatic (**C**,**D**) parameters. The semitransparent field around each curve denotes the size of the deviation from the mean value.

**Figure 8 jof-08-01112-f008:**
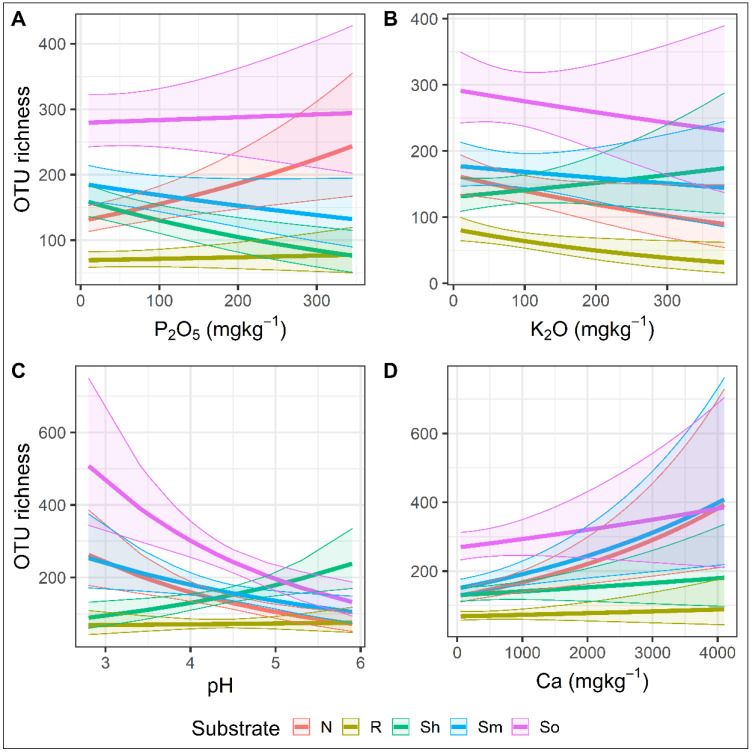
The relationship between the species richness of fungal OTUs in the different substrates (needles, shoots, roots, mineral, and organic soil) of *Pinus sylvestris* and different soil parameters (**A**–**D**). The semitransparent field around each curve denotes the size of the deviation from the mean value.

**Figure 9 jof-08-01112-f009:**
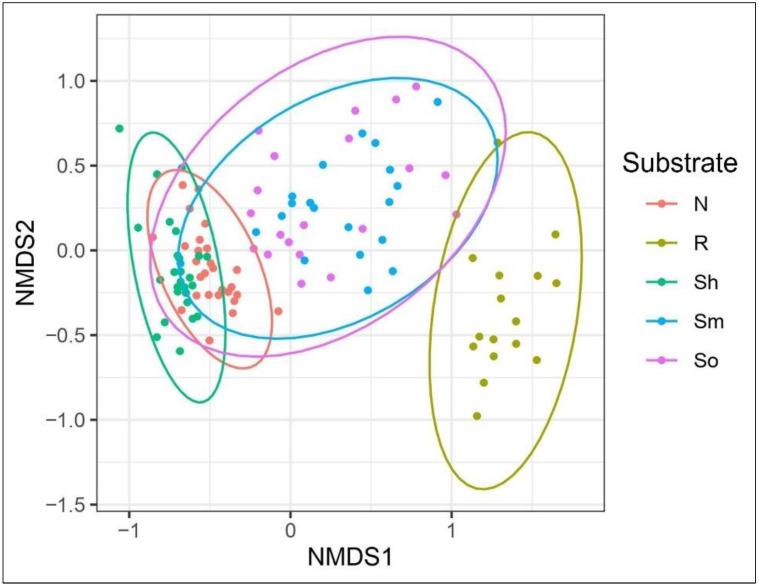
Non-metric multidimensional scaling (NMDS) of fungal communities associated with different substrates (N—needles, R—roots, Sh—shoots, Sm—mineral soil, and So—organic soil) of *Pinus sylvestris*.

**Table 1 jof-08-01112-t001:** Principal information on the 30 sampling sites in *Pinus sylvestris* forest stands.

Site	Position		Tree Species	Age	Soil Chemical Parameters							Climate Data	
No.	Longitude (N)	Latitude (E)	Composition, % *	(y)	pH, mol/L (KCl)	P_2_O_5_, mg/kg	K_2_O, mg/kg	Ca, mg/kg	Mg, mg/kg	Cl, mg/kg	Salts, ms/cm	Average Annual Temp. °C	Precipitation, mm/year
1	55°37′14″	25°11′25″	100P	60	4.0	345	25	155	48	3.6	2.55	8.0	508.8
2	53°58′52″	23°56′56″	100P	98	4.2	93	31	212	36	3.6	1.61	8.0	484.6
3	53°58′27″	24°25′48″	100P	53	4.2	123	25	256	74	3.6	1.89	8.0	484.6
4	54°8′20″	24°11′51″	100P	53	4.4	64	23	260	60	3.6	1.44	8.0	484.6
5	54°52′40″	23°42′17″	80P,20B	67	4.1	60	19	205	50	3.6	2.18	8.3	614.9
6	55°10′27″	24°26′43″	100P	57	4.0	139	19	234	55	5.3	2.41	8.3	614.9
7	55°1′10″	24°12′56″	100P	71	3.5	19	57	181	45	5.3	2.71	8.6	519.8
8	54°52′15″	23°26′41″	100P	49	3.9	18	61	298	62	3.6	3.45	8.9	491.7
9	56°11′40″	21°28′13″	100P	59	3.8	34	65	643	115	5.3	4.45	8.9	491.7
10	56°1′1″	21°6′45″	90P,10S	47	4.0	92	31	246	46	3.6	1.74	8.3	643.6
11	55°45′36″	24°41′38″	70P,20S,10B	72	3.4	13	36	191	50	3.6	3.07	8.1	479.8
12	55°45′36″	24°41′38″	80P,20B	68	5.9	71	60	4100	670	6.9	20.2	8.1	505.5
13	56°11′12″	22°23′4″	100P	58	4.2	25	86	2081	272	5.3	5.18	8.0	480.4
14	54°52′16″	25°41′56″	100P	65	4.1	41	57	339	68	3.6	2.81	8.0	755.4
15	54°34′28″	23°57′2″	100P	47	4.0	115	41	243	57	3.6	2.21	8.6	595.2
16	55°28′16″	23°26′39″	100P	55	4.2	70	39	373	66	5.3	2.08	7.8	458.0
17	55°29′54″	21°57′21″	70P,20S,10B	65	2.8	138	381	1171	271	7.1	15.9	7.7	514.6
18	56°3′26″	25°42′28″	90P,10S	59	3.9	10	51	208	41	3.6	3.02	8.4	617.0
19	55°1′46″	22°42′14″	70P,20J,10B	36	3.9	78	30	178	53	3.6	1.78	8.6	519.8
20	54°17′17″	25°39′41″	90P,10S	50	3.9	44	44	191	54	5.3	3.29	8.0	530.6
21	55°10′12″	25°41′56″	100P	65	4.4	57	13	97	28	3.6	2.62	7.4	669.2
22	55°19′38″	22°27′14″	70P,20B,10S	60	3.5	16	174	834	281	3.6	5	8.6	591.3
23	55°45′34″	21°43′12″	70P,20S,10B	65	3.4	30	83	383	81	5.3	4.29	8.0	598.7
24	54°44′58″	24°41′59″	100P	58	5.9	271	67	1624	215	3.6	8.49	7.3	614.9
25	54°56′54″	24°41′46″	100P	55	4.9	104	29	399	68	5.3	3.19	8.0	535.8
26	55°10′47″	24°42′10″	100P	50	3.9	130	36	183	52	3.6	2.72	8.0	506.9
27	55°19′1″	25°42′54″	100P	65	4.5	37	48	675	97	5.3	4.86	7.7	505.4
28	54°25′23″	24°57′33″	90P,10B	48	4.1	14	33	180	44	5.3	2.5	8.0	593.2
29	54°25′39″	24°27′15″	90P,10B	70	4.3	38	48	323	66	3.6	3.88	8.0	540.9
30	54°7′57″	23°41′50″	100P	69	5.3	62	40	1053	115	3.6	4.03	8.4	565.2

* P—*Pinus sylvestris*, S—*Picea abies*, B—*Betula pendula*, J—*Alnus glutinosa*. Tree species composition is based on volume.

**Table 2 jof-08-01112-t002:** The number of high-quality sequences and fungal OTUs from each study site.

Site	No. of High-Quality Sequences/OTUs						Shannon Diversity Index (H)					
No.	Roots	Shoots	Needles	Soil O *	Soil M *	Soil O + M	Roots	Shoots	Needles	Soil O *	Soil M *	Soil O + M
1	6569/74	271/43	1117/136	3165/283	209/77	3374/311	2.28	2.78	3.78	4.20	3.84	4.29
2	2558/117	224/59	1529/134	414/85	639/148	1053/193	2.83	3.34	3.26	3.35	4.06	4.19
3	2189/91	597/61	2518/125	1092/145	863/193	1955/279	3.21	2.61	2.99	3.58	4.57	4.34
4	1107/50	3779/134	870/61	3205/237	1962/131	5167/312	2.60	2.91	2.82	3.64	3.53	3.91
5	907/67	458/93	770/85	3180/273	742/87	3922/305	3.04	3.88	3.06	3.20	3.05	3.40
6	139/33	49/30	1846/182	236/101	390/136	626/191	3.06	3.25	4.03	4.05	4.00	4.27
7	3026/108	517/104	1974/174	1/1	21/8	22/9	2.93	3.81	3.54	-	1.59	1.70
8	2390/97	1006/125	479/76	708/142	1609/157	2317/246	3.30	3.53	3.35	4.34	3.70	4.31
9	969/88	708/119	2286/180	1427/234	1888/208	3315/322	3.18	3.84	3.75	4.47	4.02	4.42
10	1309/126	1715/222	4841/325	39/29	2407/184	2446/198	3.04	4.19	4.24	3.24	3.80	3.85
11	411/104	3032/248	1058/128	382/97	191/57	573/125	3.76	4.06	3.78	3.82	3.50	4.05
12	885/89	3134/237	689/97	1074/223	731/80	1805/253	3.18	4.03	2.58	4.66	2.93	4.37
13	75/36	2018/169	3590/188	6088/242	3955/262	10043/371	3.30	3.80	3.01	3.81	4.08	4.09
14	5076/103	1726/156	677/102	1275/267	1018/202	2293/367	2.47	3.68	3.23	4.86	4.65	5.11
15	223/56	1252/115	4097/219	1508/152	1339/137	2847/245	3.28	3.56	3.83	3.71	3.20	4.02
16	79/14	4344/221	3936/232	168/75	87/46	255/107	1.97	3.67	3.77	3.71	3.48	4.02
17	75/30	3770/194	3037/178	883/123	640/138	1523/231	3.04	3.85	3.29	3.73	4.10	4.42
18	841/51	5417/202	3649/209	113/48	285/78	398/106	1.97	3.68	3.87	3.15	3.54	3.75
19	174/50	5273/186	5146/204	27/21	152/79	179/93	3.21	3.36	3.37	2.97	3.97	4.12
20	139/38	2440/150	3122/140	1355/242	574/106	1929/289	3.00	3.43	3.28	4.81	3.86	4.84
21	121/42	913/86	2819/189	701/142	234/92	935/210	3.18	3.18	3.19	4.18	4.17	4.61
22	-/-	2368/153	4445/216	87/54	61/31	148/83	-	3.37	3.71	3.70	3.35	4.04
23	169/33	4620/233	1885/137	1047/128	1752/200	2799/286	2.56	3.67	2.86	3.47	4.40	4.54
24	219/41	2890/158	2572/148	68/16	136/35	204/45	2.65	3.89	3.39	1.84	1.65	1.88
25	2193/61	1490/155	682/89	4009/345	1721/159	5730/396	1.62	3.93	3.26	4.39	4.06	4.57
26	149/35	4207/239	397/89	1575/183	553/112	2128/245	2.52	3.95	3.66	3.57	3.37	3.74
27	1395/88	4081/159	417/71	3611/321	2120/148	5731/392	2.71	3.36	3.08	4.40	3.48	4.44
28	1868/109	652/85	1102/119	1257/166	1188/195	2445/298	3.12	3.32	3.18	3.62	4.34	4.39
29	699/58	5143/217	710/91	1960/242	2598/218	4558/364	2.58	3.47	3.48	4.56	4.21	4.67
30	-/-	10667/233	776/100	1502/166	1234/183	2736/276	-	2.78	3.42	3.83	4.13	4.27
Total	35954/734	78760/907	63031/953	42117/1440	31254/1233	73371/1854						

* O—organic layer, M—mineral layer.

**Table 3 jof-08-01112-t003:** Relative abundance (%) of fungal OTUs associated with needles, shoots, roots, and organic and mineral soil of *Pinus sylvestris*. All study sites are combined.

Fungal OTU	Phylum *	Genbank/UNITE Reference	Sequence Similarity, %	Needles, %	Shoots, %	Roots, %	Soil O **, %	Soil M **, %	Soil all, %	All, %
*Dothideomycetes* sp. 5208_5	A	KX908472	99	20.445	3.389	0.039	3.217	1.709	2.575	6.953
Unidentified sp. 5208_1	A	KP891398	100	4.314	12.786	0.017	1.363	1.654	1.487	5.530
*Archaeorhizomyces* sp. 5208_0	A	MH248043	100	1.231	0.382	8.352	8.393	9.839	9.009	4.257
Unidentified sp. 5208_2	A	MN902367	100	4.406	7.096	0.011	2.275	1.814	2.078	3.940
*Helotiales* sp. 5208_17	A	KY742593	100	1.577	6.136	0.011	0.617	0.573	0.598	2.497
*Rhinocladiella* sp. 5208_3	A	KM056296	98	0.398	6.435	0.008	0.769	0.518	0.662	2.313
*Phacidium lacerum*	A	MN588163	100	4.515	0.113	0.036	3.039	1.248	2.276	1.839
*Phialocephala fortinii*	A	MN947395	100	0.160	0.041	11.056	0.468	0.458	0.463	1.771
*Scoliciosporum umbrinum*	A	KX133008	100	0.887	4.247	0.003	0.477	0.694	0.570	1.722
*Sydowia polyspora*	A	MN900630	100	3.428	1.894	0.031	0.567	0.467	0.525	1.612
*Mycena cinerella*	B	KT900146	100	0.024	0.009	8.847	1.135	0.205	0.739	1.491
*Malassezia restricta*	B	LT854697	100	1.228	0.670	0.456	1.854	3.180	2.419	1.291
*Microsphaeropsis olivacea*	A	MT561396	100	2.588	0.819	0.111	0.803	1.641	1.160	1.261
*Penicillium camemberti*	A	MT355566	100	0.159	0.024	5.326	1.648	1.037	1.387	1.215
*Phaeomoniella pinifoliorum*	A	MK762595	100	0.982	1.769	0.006	0.558	0.483	0.526	0.956
Unidentified sp. 5208_36	A	MG828311	100	0.695	2.051	0.008	0.480	0.368	0.432	0.945
*Cladosporium herbarum*	A	MT635288	100	1.826	0.432	0.631	0.784	0.848	0.811	0.921
*Chaetothyriomycetidae* sp. 5208_56	A	KX589170	98	0.227	2.400	-	0.078	0.483	0.251	0.883
*Leptosphaeria* sp. 5208_64	A	JQ044439	97	0.252	2.417	-	0.024	0.022	0.023	0.828
Unidentified sp. 5208_39	A	MT242010	100	1.599	1.023	0.006	0.380	0.262	0.330	0.820
*Hyaloscypha variabilis*	A	MT469925	100	0.057	-	3.221	1.099	1.142	1.118	0.802
*Umbelopsis dimorpha*	Z	MT138616	100	0.114	0.004	0.028	1.885	3.209	2.449	0.749
*Chaetothyriales* sp. 5208_15	A	KP400572	100	0.836	1.346	0.019	0.297	0.381	0.333	0.732
*Trechispora* sp. 5208_19	B	JX392812	99	0.002	-	3.919	0.582	0.477	0.537	0.718
Unidentified sp. 5208_12	A	FJ553582	100	0.062	0.004	0.006	2.123	2.726	2.380	0.713
Unidentified sp. 5208_32	B	MN902363	100	2.189	0.325	-	0.252	0.070	0.174	0.702
*Cenococcum geophilum*	A	HM189724	100	0.010	-	2.517	0.560	1.817	1.096	0.683
Unidentified sp. 5208_72	A	MN902396	100	1.637	0.458	0.014	0.525	0.224	0.397	0.673
Unidentified sp. 5208_23	A	MT237078	100	1.109	0.684	0.006	0.565	0.461	0.521	0.646
Unidentified sp. 5208_69	A	MN902387	100	1.474	0.543	-	0.216	0.083	0.159	0.587

* A—Ascomycota, B—Basidiomycota, Z—Zygomycota; ** O—organic layer, M—mineral layer.

**Table 4 jof-08-01112-t004:** Pairwise comparison of the fungal communities among different substrates (needles, roots, shoots, mineral soil, and organic soil) of *Pinus sylvestris*.

Substrate	Needles			Roots			Shoots			SoilM		
	R^2^	F	*p*	R^2^	F	*p*	R^2^	F	*p*	R^2^	F	*p*
Roots	0.37	24.19	0.0011	-	-	-	-	-	-	-	-	-
Shoots	0.23	14.86	0.0011	0.36	22.06	0.0011	-	-	-	-	-	-
SoilM	0.26	15.64	0.0011	0.20	8.25	0.0011	0.25	14.50	0.0011	-	-	-
SoilO	0.21	11.88	0.0011	0.18	7.68	0.0011	0.20	11.29	0.0011	0.02	0.90	0.591

## Data Availability

Not applicable.

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
