# Peer review of "Principal Drivers of Fungal Communities Associated with Needles, Shoots, Roots and Adjacent Soil of Pinus sylvestris"

_jof, 2022, doi:10.3390/jof8101112_

Round 1
Reviewer 1 Report
The study looked into the role of climatic factors and soil chemical properties on fungal richness and how compartments structure fungal communities. The authors found interesting results and discussed them sufficiently in the manuscript. I request authors to incorporate changes related to the following queries.
· The information related to climatic (temperature/precipitation) and edaphic factor (soil pH) shaping fungal richness can be mentioned concisely in the abstract (L14-L26).
· The increase in soil pH also reduces rhizosphere priming effect, which leads to increase in the accumulation of carbon in soil, which may also be one of the reasons for decrease in fungal richness of soil. Decrease in fungal richness in relation to increase in DOC was observed by the authors of this article as well 10.3389/fmicb.2020.542220 (L263-L264).
· The reason for higher saprotroph abundance is mentioned in the manuscript as either disturbance or soil pH. There is no mention of disturbance history in site description section. Please mention any disturbance event occurred in the past or remove the disturbance related information in L449.
· Soil fungi had higher Shannon diversity index – Need to explain why fungal diversity is higher – is it related to availability of wide variety of resources? (L478)
· The possible reason behind highest percentage of unique OTUs in soil was not explained- again is it related to resource availability? (L272)
· Indexes – change to indices (L456)
Author Response
We are grateful for valuable comments and suggestions, which allowed to improve the manuscript. Please see attached our point-by-point responses to these comments and suggestions. All line numbers refer to the revised version of the manuscript. All changes in the manuscript are also highlighted by MS Word track changes.

Reviewer 2 Report
The manuscript investigated the fungal community composition of Pinus sylvestris growing under different environmental conditions using high-throughput sequencing data. More interesting, authors investigated fungal community composition and function within different habitats including needle, shoot, and root of Pinus, in addition to mineral and organic soil samples which were collected at 30 sites. Using correlation analysis between environmental variables and soil chemical properties in relation to fungal biodiversity, this study confirmed that a combination of environmental factors and stand characteristics significantly shaped the fungal diversity of phyllo/endo/or rhizosphere.
Minor comments:
GenBank accession numbers should be provided in L212.
The assessment of fungal functional groups which showed the most fungal OTUs (Fig. 6) need to be improved to be more informative for differentiation between diverse habitats.
Result section contain 4 tables and 9 Figs. Authors can shorten this section, for example by moving Table 2 and 4 into the supplementary data.
L 128, “with P. sylvestris what can” aching into….” with P. sylvestris which can”.
L 133, check the number 500 m2 “which were 500 m2 in size, situated”.
Author Response

(The authors gave the same response as above.)

Round 2
Reviewer 1 Report
I am happy with revised version of the manuscript and it can be accepted in its present form.